# Dynamic Topology Reconfiguration of Boltzmann Machines on Quantum Annealers

**DOI:** 10.3390/e22111202

**Published:** 2020-10-24

**Authors:** Jeremy Liu, Ke-Thia Yao, Federico Spedalieri

**Affiliations:** 1Department of Computer Science, University of Southern California, Los Angeles, CA 90007, USA; 2Information Sciences Institute, University of Southern California, Marina del Rey, CA 90292, USA; kyao@isi.edu (K.-T.Y.); fspedali@isi.edu (F.S.); 3Department of Electrical and Computer Engineering, University of Southern California, Los Angeles, CA 90007, USA

**Keywords:** quantum annealing, Boltzmann machines, machine learning, entropy, algorithms

## Abstract

Boltzmann machines have useful roles in deep learning applications, such as generative data modeling, initializing weights for other types of networks, or extracting efficient representations from high-dimensional data. Most Boltzmann machines use restricted topologies that exclude looping connectivity, as such connectivity creates complex distributions that are difficult to sample. We have used an open-system quantum annealer to sample from complex distributions and implement Boltzmann machines with looping connectivity. Further, we have created policies mapping Boltzmann machine variables to the quantum bits of an annealer. These policies, based on correlation and entropy metrics, dynamically reconfigure the topology of Boltzmann machines during training and improve performance.

## 1. Introduction

We present an information-theoretic approach for reconfiguring Boltzmann machine (BM) connectivity to improve performance when implemented on D-Wave’s open-system quantum annealing hardware. BMs typically feature a restricted bipartite connectivity to make training tractable, but annealing hardware allows us to implement BMs with expanded connectivity. However, the physical constraints of the hardware only allow us to implement a small subset of possible expanded connectivity options, so we explore methods for choosing subsets that are more likely to be beneficial (Though this paper uses quantum annealing hardware, we caution that we do not address issues of quantum supremacy or speedup. Quantum algorithms have been proven to be superior to classical algorithms in certain areas but have yet to find quantum hardware capable of implementing them. There has been a notable recent finding from Google showing evidence of quantum supremacy [1]. Although the specific application found within that paper is not generally useful for quantum supremacy, it is encouraging that quantum supremacy, in some form, has been empirically observed.). Our implementation of Boltzmann machines on quantum annealers differs from those typically used by other researchers. Some choose to implement full Boltzmann machines that represent both visible and hidden units on the annealer [2,3], while some discuss connectivity topologies but do not implement them on hardware [4,5]. The Boltzmann machines we implement on quantum annealers map only the hidden units and their intra-layer connections to hardware, avoiding the difficult subgraph isomorphism problem found in the other approaches. We also introduce the idea of dynamic reconfiguration of BM topology in which we change our mappings during training to improve performance. Mapping choices determine BM connectivity topology because annealer qubits and couplers are fixed in place while we can freely change which qubits represent which BM variables. Our results show that mapping decisions designed to lower entropy in BM topology lead to better outcomes.

In Section 2, we review Boltzmann machines, discuss how we modify them for implementation on quantum annealers, and show that different mappings of BM variables to hardware can produce better results. In Section 3, we look at correlation, entropy, and total correlation as methods for designing better qubit mappings. Section 4 shows how we dynamically alter mappings during training to improve performance. In Section 5, we offer an explanation of our results using the concept of hidden information encoding.

## 2. Mapping Boltzmann Machines to Quantum Annealers

A Boltzmann Machine (BM) is a probabilistic generative model defined over a set of *N*-bit binary strings. Each network state has an energy given by
(1)$$E\left(x\right)=-\left(\sum_{i<j}w_{ij}x_{i}x_{j}+\sum_{i}\theta_{i}x_{i}\right)$$
where the weights $w_{ij}$ and $\theta_{i}$ are the parameters of the model. The probability a given state is sampled from the network is given by $P\left(x\right)=e^{-E(x)}/Z$, where $Z=\sum_{x}e^{-E(x)}$ is a normalization factor known as the partition function. Training a BM corresponds to computing the values of the weights that would make the model reproduce the empirical distribution characterizing a set of data points. A Restricted Boltzmann machine (RBM) is a variant of the BM in which network connectivity is restricted to that of a fully bipartite graph. One partition contains visible units, the other contains hidden units, and only visible-hidden connectivity is allowed.

For the RBM the energy function takes the form
(2)$$E\left(v,h\right)=-\left(\sum_{i,j}w_{ij}h_{i}v_{j}+\sum_{i}\theta_{i}v_{i}+\sum_{j}\xi_{j}h_{j}\right)$$

Training is performed by minimizing the log-likelihood of the data set over the parameters of the model (the weights $w_{ij}$ and the biases $\theta_{i},\xi_{j}$). Setting the gradient of the log-likelihood to zero results in the well-known update rule for the weights [6], which is proportional to the difference between expected values over the data set and over the model, i.e.,
(3)$$\Delta w_{ij}\propto {\langle v_{i}h_{j}\rangle }_{data}-{\langle v_{i}h_{j}\rangle }_{model}$$with similar expressions for the biases. Computing the expected value over the data set is straightforward, but the average over the model is computationally expensive to obtain. Approximate algorithms such as contrastive divergence (CD) [7] have been developed to accelerate the computation of this term, and any approach that can approximate this term has the potential to accelerate the training of RBMs and BMs. Quantum annealing can be used to approximate this quantity and integrate this new technology with machine learning.

### 2.1. Limited Boltzmann Machines

RBMs normally enforce bipartite connectivity, meaning the only connections allowed are between hidden units and visible units. When we ease this restriction to allow limited connections to exist among hidden units, we call this semi-restricted topology a limited Boltzmann machine (LBM). LBMs can be viewed as a superset of RBMs, the only difference being a set of extra available connections between hidden units.

LBMs share the same problem fully connected Boltzmann machines face in that exact inference cannot be used to calculate the distribution of hidden unit states. Conditional independence, which RBMs rely upon, is broken by the LBM’s connectivity structure, making efficient training and evaluation difficult. However, we do have a sampling device at our disposal: an open-system quantum annealer that solves the sort of optimization problem Boltzmann machines represent. In place of using exact inference (Equation (3)), we can instead use annealing hardware to draw samples from a complex distribution. The rest of the contrastive divergence training procedure remains the same.

Figure 1 shows a LBM that features connectivity between hidden units that conforms to the physical limitations of our quantum annealing hardware. It also illustrates how a quantum annealer helps us carry out contrastive divergence training in LBMs.

We do not use full Boltzmann machines because we have a limited number of qubits and connections between qubits (couplers) on the annealing hardware. To deal with the qubit constraint, our implementation of LBMs on a quantum annealer (later in Section 2.3) does not use qubits to represent our data set, i.e., the visible layer is not encoded in the quantum annealer. Representing full Boltzmann machines on the annealer would only allow us to solve problems of trivial size. Our usage of approximate training via contrastive divergence allows us to use qubits to represent only the hidden units and the connections between them on an annealer.

### 2.2. Data

Our chosen data sets are the MNIST handwritten digits data set and a MoS_2_ thermalization data set. Our first several benchmark plots show results derived from the MNIST data set, and later plots extend our experiments to include results from MoS_2_.

The MNIST data set is a collection of hand-written digits extensively studied in the deep learning community. The data set contains several thousand images of size 28×28=784 pixels. (The size of the data is not a strict limitation on our experimental design because we do not map visible units (representing data) onto qubits. Rather, we map hidden units to qubits, and we have total control over the number of hidden units we choose to use. We chose the data set to iterate our design over and develop our approach.).

The second data set we use is a MoS_2_ thermalization data set. MoS_2_ is a molybdenum-sulfide monolayer that produces a variety of different structures when heated and rapidly cooled. The original 3-dimensional data set was composed of 4.3 million atoms in a 211.0×96.3×14.5 nm^3^ space, and a subset was sliced into 5000 2-dimensional image slices.

### 2.3. Implementing a Boltzmann Machine on D-Wave

The machine we use, D-Wave’s open-system quantum annealer, is located at the USC-Lockheed Martin Quantum Computing Center. The processor represents an Ising Hamiltonian using an array of superconducting flux qubits with programmable interactions [8]. The qubits are implemented using Superconducting QUantum Interference Devices (SQUIDs) composed of elongated Niobium loops. The loops and Josephson junctions control quantum annealing and compensate for slight differences between the physical properties of any two SQUIDs due to fabrication variations. The qubits are divided into 8-unit chimera cells, which have four qubits arranged horizontally and four vertically such that each qubit intersects orthogonal qubits. At these intersections a SQUID is placed to control the magnetic coupling between the corresponding horizontal and vertical qubits within the same cell. This architecture results in a coupling graph that is fully bipartite at the cell level. The processor is composed of cells joined together in a square lattice such that the horizontal qubits in one cell are coupled to the horizontal qubits in the neighboring cells to the right and the left, and the vertical qubits are coupled to the vertical qubits on top and on the bottom. This creates a (M,L) Chimera graph ((M,L) Chimera graphs have M×M cells and *L*-sized partitions with 2L units per cell. The DW2X implements a (12,4) Chimera graph with 1098 working qubits.).

Deep learning using BMs has been proposed before, but as previously discussed, learning is intractable for fully connected topologies because we need to compute expected values over an exponentially large state space [9,10]. RBMs address this by restricting network topology to bipartite connectivity to introduce conditional independence among visible units (representing the dataset and RBM output) given the hidden units (representing latent factors that control the data distribution), and vice versa, though they lose some representational power in the process. Quantum annealing hardware gives us an opportunity to train LBMs using its unique sampling properties and compare their performance against standard RBMs to investigate how more interconnected topologies improve results.

Our RBM has 784 visible units to represent each pixel in a 28×28 MNIST digit image and 80 hidden units on a D-Wave quantum annealer. Similarly, we used 32×32=1024 visible units to represent the MoS_2_ data and 80 hidden units. The BMs were trained over 200 epochs on a training set and evaluated against a test set. After establishing the baseline performance of RBMs, we loosened the bipartite topology restriction and began comparing RBM performance to LBM performance.

Because D-Wave hardware has a physical constraint on the number of qubits and connections, we would have to solve an embedding problem to represent a full BM. This requires a large overhead of qubits, which means we can only use small problems. However, if we choose to use LBMs instead, we can avoid the embedding issue. We represent only the hidden units and continue with contrastive divergence using the annealer as a sampler. LBMs are also convenient because we can directly control how many hidden units we want to use. Using full BMs requires representing each data dimension using a qubit whereas our approach with LBMs allows us to represent the data off the chip, letting us use all the qubits to represent the hidden units. Figure 2 shows how we use one chimera cell to represent a portion of the hidden layer.

Using D-Wave hardware to optimize LBM parameters addresses the training intractability issue because the quantum annealer does not rely on conditional independence between units within a layer. We review the training process for BMs to illustrate.

The configuration of binary states *s* of units has an energy *E* defined by
(4)$$E\left(s\right)=-\sum_{i}s_{i}b_{i}-\sum_{i<j}s_{i}s_{j}w_{ij}$$
where *b* is the bias of a unit and $w_{ij}$ is the mutual weight between two units *i* and *j*. The partition function is $\sum_{s}e^{-E(s)}$, and the probability the BM produces a particular configuration *x* is
(5)$$P\left(x\right)=e^{-E(x)}/\sum_{s}e^{-E(s)}.$$

P(x) is difficult to compute in a full BM because it requires a sum over an exponentially large state space. If we want to determine the probability of some hidden unit *i* is on (equal to 1) without any guarantee of conditional independence, we would have to calculate $P\left(h_{i}=1\right)=P\left(h_{i}=1|v,h_{-i}\right)$, where *v* is the state configuration of visible units and *h* is state configuration of the hidden units. However, if we use RBMs to restrict ourselves to bipartite connectivity between *v* and *h*, this probability factorizes and we can write $P\left(h_{i}=1\right)=\prod_{j=1}^{n}P\left(h_{i}=1|v_{j}\right)$. Our first RBM baseline experiment uses this standard procedure with 1-step Gibbs sampling to analytically find the distribution of unit states. In our LBM experiment, instead of using conditional independence or Gibbs sampling to analytically find the state distribution, we use quantum annealing to approximate the state distribution.

Earlier, we mentioned we only represent the hidden units on the annealer. This is done by reducing the influence of visible units on the hidden unit state distribution to a simple bias on the qubit representing a given hidden unit. Suppose we want to sample from the annealer to get an empirical measure of the hidden unit state distribution given the visible unit states as a part of our contrastive divergence training algorithm. The distribution of hidden unit states is defined by Equation (5). Critically, one part of E(x) is fixed: the aforementioned visible unit states. If we look at the latter term of Equation (4), $\sum_{i<j}s_{i}s_{j}w_{ij}$, we notice that any of those two-body interactions involving a visible unit will be fixed. There are two possibilities: in the first, a hidden unit configuration excludes such an interaction, in which case we are done. In the second case, a hidden unit configuration includes such an interaction, in which case $s_{i}s_{j}w_{ij}$ is contributed to the overall energy of that state. In effect, this is simply a bias on the hidden unit since we already know the visible unit state involved in all these interactions. This observation allows us to represent only the hidden units on the annealer.

The training procedure for LBMs is the same as for RBMs, except that we augment Equation (3) to no longer explicitly include only interactions between visible and hidden units:(6)$$\Delta w_{ij}\propto {\langle s_{i}s_{j}\rangle }_{data}-{\langle s_{i}s_{j}\rangle }_{model}$$

We also replace Gibbs sampling with a call to the annealing hardware, collecting empirical samples to approximate a distribution of hidden unit states.

The weight matrices we use are randomly initialized from a standard normal distribution and updated using the rule in Equation (6).

### 2.4. Impact of Qubit Mapping and Spatial Locality

Prior work [11] has shown that LBMs perform better than RBMs on the same data sets. Here, we are interested in how our choice of mapping hidden units to hardware qubits affects the performance of BMs.

Our initial experiments with LBMs used a 1:1 mapping of BM hidden units to hardware qubits according to the order in which we defined them. This was done to keep implementation straightforward. However, there are many other ways we could have chosen to map hidden units to qubits. One thought is to map in such a way as to exploit data locality, especially as we use image data sets. We expanded our straightforward initial mapping to include spatial locality considerations.

To emphasize the impact of hidden-to-hidden connections in our LBMs, we deleted the majority of visible-to-hidden connections such that each hidden unit could only see a 4×4 pixel square of image pixels.

In this set-up, each hidden unit is associated with a specific area of the image. To take advantage of data locality, instead of simply assigning hidden units to qubits in the order they were defined, we put hidden units representing contiguous areas of the image together in the same chimera cell so the units could exchange relevant local information with each other. We call one such mapping a “box” mapping because the qubits we place together in a cell represent a square section of the image. Similarly, a “line” mapping represents an image-width line of pixels.

We compared our box and line mappings against a randomly generated assortment of other mappings. As expected, the box mapping demonstrated improvement over the standard 1:1 mapping, but we also saw other mappings that performed better still. Figure 3 shows early results demonstrating that better mappings than the box mapping exist.

## 3. Finding Better Qubit Mappings

Previous results show that some mappings perform better than others. Given that better mappings exist, we want to find them. Though using locality-exploiting mappings was a good first step that yielded better results, what we seek is a generalized approach for creating good qubit mappings and a mathematical explanation for our results.

We start by discussing static qubit mappings—mappings that are determined early in training and remain unchanged afterwards. Section 3.1 describes how we utilized correlation to design better mappings in a more objective manner. Section 4 moves on to examine dynamic qubit mappings—mappings that are evaluated and changed throughout training—and how we used entropy to create better maps. In Section 5, we offer a mathematical explanation for how entropy influences qubit mapping quality and draw connections to similar work conducted by other researchers.

We would like to stress that in terms of using correlation, our approach is meant to understand if any correlations present in the output of the D-Wave device can be leveraged to improve the overall performance, without delving into the deeper subject of whether these correlations are quantum (i.e., related to entanglement between qubits) or merely classical. That is more a question about the physics of the D-Wave device, and it is outside the scope of the present work. It has already been demonstrated experimentally that the D-Wave processor is indeed capable of producing entangled states [12], but how that entanglement could improve any computational task is still an open and widely researched topic at the frontier of quantum computing.

### 3.1. Correlation

The creation of line and box qubit mappings by hand was made possible by domain knowledge of the data set, namely, that spatial locality is important. However, we want to go beyond manual creation of qubit mappings and develop a generalized approach. Our next step was to examine correlations between qubit activity. We eventually concluded that maximization of correlation within our chimera cells produces the best results.

To calculate how qubits were correlated with each other based on activation, we recorded the hidden states of a Boltzmann machine induced by the BM’s exposure to our input data patterns. This record of hidden unit states was the basis for generating our matrix of qubit correlation values. We used Pearson’s correlation coefficient, which ranges within [−1,1].

We found that as BMs trained, correlation among hidden units trended weaker and tended to settle at stable values. We interpreted this weak correlation as a desirable state for the BM as such conditions produced the best results.

Given this interpretation that weak correlation among hidden units might improve results, we wanted to design a qubit assignment policy based on correlation coefficients. A BM with *k* hidden units has $O(k^{2})$ correlation coefficients, but the quantum annealer only has O(k) couplers available to enable interaction between hidden units, due to the restricted connectivity of the Chimera graph. In order to implement a BM on annealing hardware we would be forced to choose a subset of interactions to enable, therefore we should choose to enable the most beneficial interactions. Combined with our observation that BM performance seems to influence or be influenced by correlation magnitude, we created the following remapping procedure.

Find an available (unassigned) pair of qubits that have the highest correlation coefficient.Place the qubits in the opposing partitions of an unused chimera cell.For each of these “seed” qubits, find the 3 other qubits most correlated with the seed and place them in the opposing partition.Repeat until all qubits and chimera cells are assigned.

We started with a RBM and partially trained it for 10 epochs; we chose 10 epochs due to our observation that qubit correlations seem to settle around that time for our choice of hyperparameters on the MNIST digit data set. After performing this weak remapping procedure, we then converted the RBM to a LBM (Conversion of RBM to LBM is straightforward. Recall that LBM connectivity is a superset of RBM connectivity, so all we need to do is add previously non-existent couplers to our BM. The only thing a mapping affects is our choice of which hidden-to-hidden connections (many) get the chance to be represented by a physical hardware couplers (few).) using the new mapping and continued training. Our results showed that our efforts created a performance improvement.

Figure 4 shows a plot comparing the performances (The figure shows that minimizing correlation, the “minCorr” policy, appears to work best, whereas we had originally stated that maximizing correlation was the better choice. Both are true, in a fashion. Minimizing correlation works the best when we have only a few (less than 50) epochs of training. However, as we will see in a later experiment, extending training beyond 50 epochs allows a maximal correlation policy to eventually catch up and surpass a minimum correlation policy.) of different qubit mappings. As we had guessed, attempts to influence the correlation between qubits within a chimera cell did alter the performances of the Boltzmann machine.

In addition to the mapping procedure we just described, we also developed a variant policy and tested it on a separate data set, the MoS_2_ thermalization data set, to both check if our results held up across data sets and to have another policy to compare against.

The variant of our remapping policy is shown in Figure 5. We now describe this alternate policy, which we call a “shoelace” policy:Find an available pair of qubits, **x** and **y**, that have the highest correlation coefficient.Place **x** and **y** in the opposing partitions of an unassigned chimera cell.Find the qubit with the greatest correlation with **x** and place it in the partition opposite of **x**; this qubit is the new **x**. Do the same with **y**.Repeat step 3 until the chimera cell is full.Repeat until all qubits and chimera cells are assigned.

This alternate policy did not perform as well as our original policy, but we created it primarily as a comparison point. It, alongside comparisons to box and line mappings, reinforced our belief that qubit mapping policies do have a significant effect on the eventual training outcome of Boltzmann machines. Although these correlation-based qubit mapping policies were a nice starting point for us, we recognized there were more sophisticated approaches available.

Figure 5 contains the results from our experiments on the MoS_2_ data set. The BMs were trained for 10 epochs before qubits were remapped according to different policies. Notably, we had a remapping policy that beat out the choice of forgoing remapping altogether. Up until this experiment we had been mostly concerned with how policies such as box mapping, line mapping, random mapping, or weak mapping compared against each other. With this experiment, we also included the baseline experiment of choosing not to remap any qubits and have the logical qubit indices be the same as the hardware indices. The results suggest that using correlation to guide qubit remapping policies is capable of producing better outcomes for Boltzmann machines.

## 4. Dynamic Qubit Remapping With Entropy

We have used correlation to develop our first policies, but the policy objectives are still and ambiguous. We have a vague idea that we want to influence correlation within a BM, but we never developed a way to objectively measure “how much” correlation was in our network or how it was reduced. Even so, the approach with correlation coefficients showed we could design better policies using some metric, and this section considers entropy as that metric.

We also want to adjust our qubit remapping workflow to extend static qubit mappings to dynamic qubit remapping. When conducting our correlation experiments, we trained a RBM for a few epochs, calculated the correlation coefficients, created a static qubit mapping, and resumed training as a LBM. What we envision instead is for a BM to periodically reevaluate and alter its qubit mapping before committing to a final mapping. By allowing multiple reevaluations, we might find opportunities to move to better qubit mappings.

### 4.1. Calculating Entropy

When considering the entropy of hidden unit activity patterns with Boltzmann machines, we find it convenient to divide the hidden units into digestible subgroups of 4 units. If we consider all *n* qubits, we have $2^{n}$ possible activity patterns and would need to gather an appropriately large amount of data to ensure we adequately cover the space. We instead approximate the overall entropy of all the units by calculating the observed entropy for each chimera cell (composed of 2 4-unit subgroups) and summing over all chimera cells. Another reason we divide into groups of 4 is Hinton et al.’s work on capsules and routing [13] and ver Steeg, et al.’s work on correlation explanation [14], which group together computational units and manipulate their connectivity using information theoretic approaches. Hinton’s work can be seen as a means of encouraging encodings that are more amenable to human interpretation and ver Steeg’s work as a means to better optimize the explanatory power of a network. In particular, the latter work concerns itself with how to create latent variables and group them together using a total correlation metric to achieve better results; broadly speaking, a Boltzmann machine has a similar concept in its hidden layer, which is a collection of latent variables that explains the data’s distribution. Though these works are not directly applicable to our work here with Boltzmann machines, they are nonetheless conceptually motivating and influence us to use 4-unit subgroups corresponding to partitions within chimera cells.

As stated, we calculate entropy on a per-cell basis. Recall that each chimera cell is bipartite in connectivity structure, where each partition is composed of 4 qubits. Supposing that we are given a list of observed hidden unit activities within this chimera cell, it is straightforward to calculate entropy, defined as
(7)$$H\left(X\right)=-\sum_{x\in X}P\left(x\right)\log P\left(x\right)$$
where *X* is the distribution of hidden unit states. We obtain P(x) for all x∈X by counting the frequency of each activation pattern in the distribution. To obtain the distribution, we expose the BM to the data set and record the resulting non-deterministic hidden unit activation pattern (We could generate more than one encoding per input pattern due to the probabilistic nature of the Boltzmann machine, but we chose to use only one for convenience’s sake.).

Using Equation (7) can be troublesome if we have many dimensions in our distribution. Dividing the hidden unit activation distribution into chunks of 2×4-unit subgroups mitigates this issue, which also benefits scaling efforts as future hardware is likely to add more chimera cells. The new D-Wave architecture, Pegasus, features expanded connectivity but still keeps chimera cells as a basic subgraph composing the overall structure. With 8 qubits in a chimera cell, we only need to cover $2^{8}$ possible activity patterns. To summarize, for each chimera cell, we filter out all qubits except the ones present within that cell. We then apply Equation (7) to the filtered hidden unit activity patterns to obtain some entropy value. We repeat this for every cell and sum all partial entropy values to obtain a final result.

Using 4-unit subgroups also allows us to mix and match partitions of chimera cells to manipulate our entropy numbers. We will be swapping partitions around; see Figure 6 for a visual interpretation. If we try to mix and match individual qubits, we have a factorial number of possibilities to consider. Using 4-unit chimera cell partitions as a quantum drastically reduces the accounting we need to perform.

One more simplification we make is to consider only couplers that exist within a chimera cell. Within a chimera cell there are 4×4=16 couplers. However, the cell also has connections to adjacent cells—each partition has 4 such connections for a total of 8 per chimera cell. We ignore these adjacent connections to make our mapping efforts easier, but will later address these adjacent connections and suggest how we can adapt our work to include them.

With our assumptions explained, we now explain the methods we use to generate qubit mappings.

### 4.2. Dynamic Remapping

We now present a method to use our entropy metric to guide qubit mapping efforts. In general we have found that high entropy in hidden unit activity should be avoided if possible. On the other hand, while low entropy is generally desirable, achieving minimal entropy can lead to negative repercussions. We describe two methods we used to create qubit mappings of varying entropy in hidden unit activity. The first is a simple greedy method which seems to perform best with low entropy mappings, and the second is a more complex greedy method that, while producing lower entropy configurations, performs slightly worse.

We also move from static maps to dynamic remapping described earlier. One benefit of this dynamic process is that it opens up an entirely new portion of parameter space for the Boltzmann machine to explore. Previously, with a locked, static qubit mapping, a given qubit was always bound together with the same set of (up to) 6 qubits due to the physical constraints of the D-Wave quantum annealer. Now, however, the ability to remap qubits throughout the training process means that a given logical qubit has many opportunities to interact with qubits it would normally never be able to communicate with.

Figure 7 shows visually what we describe regarding the expansion of parameter space. If we have *N* qubits to represent our *N* hidden units, we potentially have $O(N^{2})$ connections among hidden units, or couplers, that we could tweak and explore. However, the D-Wave quantum annealer can only support 6 connections per qubit. A static mapping method would lock us into choosing which 6 connections out of those $O(N^{2})$ possibilities to use for each qubit and then make us commit to the decision, all prior to any training, feedback, or intermediate results. Using a dynamic method allows us to abandon poorly performing couplings to seek more promising combinations, enlarging the space we are allowed to explore.

### 4.3. Greedy Entropy Mapping

The first method we use is a greedy, non-optimal, and simple method. We can variably choose to maximize or minimize entropy; our empirical results show we want to minimize entropy to generate better results overall.

The simple greedy method is straightforward. As we are concerned with fitting quanta of 4-unit subgroups together into configurations that minimize entropy, we generate all possible pairs of 4-unit subgroups (recalling that each chimera cell can only fit two such groups), calculate the resulting pairing entropy values, and sort them accordingly. We then take the lowest entropy pairings and assign them to a chimera cell, skipping pairings if we have already previously mapped an involved subgroup to a chimera cell.

Supposing we have *N* chimera cells, we then have 2N subgroups and $\frac{2N(2N-1)}{2}=O\left(N^{2}\right)$ possible subgroup pairings. The entropy calculation for each chimera cell is constant in complexity, and sorting is O(MlogM). In our case, $M=\frac{2N(2N-1)}{2}⟹O\left(N^{2}\log N\right)$. For our annealing hardware, *N* is small. The systems we use have either N=144 or N=256 for 1k or 2k qubits, respectively. Complexity of the remapping operation is not an overriding concern because it is only executed once per training epoch at most. In our experiments, we only execute it every 5th training epoch, further minimizing the impact.

Settling on a 4-unit quantum is more manageable than focusing on individual qubits. With 4-unit subgroups we take advantage of the bipartite connectivity of chimera cells to simplify our calculations. Each chimera cell only has one way to fit two subgroups together, and it makes no difference which partition of the cell is assigned to which subgroup (There are a total of two ways to fit 4-unit subgroups into a chimera cell, but they are equivalent because couplers are symmetric connections between qubits and because we ignore inter-cell couplers.). However, had we chosen to manipulate connectivity on the individual qubit level, the ordering of qubits within a cell becomes an additional layer of concern and the number of possible configurations rapidly grows, as does the complexity of the remapping operation. A single chimera cell has 84 ways to divide its qubits into two partitions in contrast to the one choice when we use 4-qubit subgroups.

Exhaustive search of all possible variable-to-qubit assignments is not viable because there exist (8N)! possibilities for *N* chimera cells. Even using simplified 4-unit subgroups and ignoring inter-cell couplers leaves us $\frac{(2N)!}{N!2^{N}}$ possibilities. Using N=16 in our small experiments already has too many possibilities to handle, so we must choose more intelligent way to create mappings. Having shown a simple greedy method, we next describe a greedier variation.

### 4.4. Greedier Entropy Mapping

The simple greedy mapping method generates low entropy mappings but typically does not find lowest possible entropy mapping. To determine how LBMs would perform using even lower entropy configuration (under our imposed constraints), we developed a greedier algorithm to find lower entropy mappings.

Table 1 shows our greedy and greedier methods compared against each other on the MNIST data set. We trained a Boltzmann machine that remapped qubits every 5 epochs. Each remapping used the greedy method, but we also recorded the mappings generated by the greedier method for comparison purposes (a what-if scenario). Our greedier method produces lower entropy values the simple greedy method as intended. We note here that our greedier method does not produce the globally minimal entropy mapping, which we will discuss later.

The greedier algorithm uses memoization along *N*, the index of the highest-indexed chimera cell we will consider. Let us have a function MinE(n) find the minimal entropy mapping and its corresponding entropy value for chimera cells 0…n−1. We call the minimum entropy value E(n) and the mapping configuration M(n). MinE(n) will only use the hidden units currently mapped to chimera cells 0…n−1. The qubit mapping of the Boltzmann machine is only changed when we finally finish our calculation of MinE(N).

First, consider the base case where we consider only the first-indexed chimera cell, or MinE(1). There is only one single way to create a mapping, which is simply leaving the subgroups where they are. Calculating E(1) and M(1) is trivial because our mapping is already determined.

To calculate MinE(n), we can use MinE(n−1), E(n−1), and M(n−1). We already have some optimal mapping for chimera cells 0…n−2 and their associated qubits. Now, we are considering adding chimera cell n−1 and the two subgroups of qubits associated with it to our overall solution. Let us call those subgroups **a** and **b**. We are confronted with three possible solutions for MinE(n):1.Accept both **a** and **b** as a pair in the new chimera cell. It is straightforward to update E(n) and M(n). This step is overall O(1) because we do not need to iterate over anything and we do not execute any expensive functions or searches.2.Keep **a** in the new chimera cell and force **b** to reside in one of chimera cells 0…n−2. In this case, we have to run through the mapping produced by M(n−1) and try replacing each subgroup with **b** and recalculating the entropy associated with the affected chimera cell. We also calculate and store the entropy that results from the replaced subgroup being placed in the new chimera cell at index n−1. After running through the entire mapping replacing different subgroups with **b**, we choose the configuration that produces the overall lowest entropy (remembering to account for **a** being paired with a replaced subgroup in chimera cell n−1) and storing it as a potential solution to E(n), M(n), and MinE(n) accordingly. Recall that calculating entropy is a O(1) operation and that the mappings contain O(n) subgroups, so overall this step is also O(n).This step is why we call this algorithm greedier instead of optimal. Our algorithm assumes that when we force **b** to reside in an older cell 0…n−2, the subgroup that **b** replaces (call it **d**) automatically gets paired with **a**. This is not necessarily the case. It is possible that **d** can go reside in another older cell 0…n−2 and produce an overall lower entropy value and that **e**, some low-indexed subgroup, can eventually get paired with **a**. The answer may not always be optimal, but in practice in produces lower entropy than the simple greedy method.Without any loss of generality, we can also force **b** to stay in the chimera cell at index n−1 and force **a** to reside in one of chimera cells 0…n−2.3.We force **a** and **b** to reside outside of chimera cell n−1. This may seem worrying at first because we might have to calculate the entropy of all possible pairings of subgroups again, driving up the complexity of our calculation of MinE(n). However, all scenarios in this case will degenerate into cases 1 or 2. Consider two situations:(a)Suppose the cell n−1 contains neither **a** nor **b**. Furthermore, suppose **a** and **b** end up paired together in cell *i* such that 0≤i≤n−2. This is equivalent to 1 because we have assumed that inter-cell couplers are disabled. This assumption allows us to consider chimera cells independently of each other, thus their ordering does not matter. We could take cell *i* and re-index it as cell n−2, and vice versa. This degenerates into case 1 and we have no additional work to do.(b)Suppose cell n−1 contains neither **a** nor **b**. Furthermore, suppose **a** and **b** are found in different cells *i* and *j*, respectively, such that 0≤i,j≤n−2. This is equivalent to 2, again owing to our assumption that inter-cell couplers are disabled, allowing us to re-index chimera cells as we please.

After all this, we are presented with possible mappings from cases 1 and 2, recalling that case 3 degenerates into either 1 or 2. From these possibilities, we simply pick the lowest entropy mapping and update MinE(n), E(n), and M(n). The most complex operation in all these cases comes from (2), which we showed was O(n). If we wish to calculate MinE(N), we have calls to MinE(N), MinE(N−1), *…*MinE(1), so overall our complexity is $O(N^{2})$, which is actually slightly better than the $O(N^{2}\log N)$ of the greedy method.

Having established the base case of MinE(1) and shown how to calculate MinE(n) from the results of MinE(n−1), we can compute MinE for any value of *n* where *n* is the number of chimera cells that compose our Boltzmann machine.

### 4.5. Optimal Entropy

The greedier method described in Section 4.4 is not an optimal method. The algorithm to find the globally lowest entropy value can be mapped onto the minimum weight matching problem. In the minimum weight matching problem, we find an independent edge set (a set of edge with no common vertices) such that the sum of weights is minimized. Mapping our subgroup problem to this weight matching problem is easy: the vertices *V* of the graph are simply our subgroups and the edges *E* are the resulting entropy values when we pair subgroups together. The result of the weight matching algorithm corresponds to our entropy value.

The weight matching problem is $O(|V|E^{2})$ in complexity, or $O(N^{3})$ for our purposes when considering *N* chimera cells. Faster algorithms [15] improve the running time to $O(\sqrt{V}E)$, or equivalently $O(N^{2.5})$. A randomized algorithm that uses fast matrix multiplication [16] further reduces the running time to $O(V^{2.376})$.

### 4.6. Total Correlation

Our use of entropy as a metric to guide qubit remapping policies also leads us to consider the use of total correlation. Total correlation is defined as
(8)$$TC\left(G\right)=\left(\sum_{g\in G}H\left(g\right)\right)-H\left(G\right)$$

Here, g∈G is a random variable in a group *G* of random variables. Total correlation measures how much information is shared among the variables. Minimization of total correlation (TC) is an explicit goal in CorEx [14], and as Boltzmann machines can also measure TC using 4-qubit subgroups as each *G*, we can use it as an alternate metric to determine remapping policies. The only change required for implementation is to replace each H(G) calculation (see Equation (7)) with Equation (8). We found usage of TC as a metric was not particularly helpful, however, being outperformed by the simpler entropy metric. We nevertheless include TC results with our other findings for completeness.

### 4.7. Results

Figure 8 plots the reconstruction losses of various BMs trained with different remapping policies. Remapping occurred after epochs 0, 5, 10, and 15, after which there was no more qubit remapping. Training continued until epoch 200. From the results, we conclude that a policy that reduces entropy, but does not minimize it completely, performs best. We also observe that although we remap qubits early in training, the effects of remapping persist long after dynamic remapping ceases. Of particular note is the amount of training time saved. Directly comparing loss values against each other is not illuminating because we do not know the lower bound on L2 achievable by the BMs; however, what we can do is compare the number of epochs necessary for each qubit remapping policy to match the others. For instance, the loss value obtained by the no-remapping policy at epoch 200 is the value achieved by the greedy entropy policy by epoch 160—the greedy entropy only took $\frac{3}{4}$ the training time to get the same level of performance. This result is attributable to remapping changes we made early in training. Our choice of qubit remappings in the first 15 epochs (totaling only 4 remappings) persisted through the entirety of training and led to a disproportionately large performance gap between policies. No qubits were added or removed and no hyperparameters were altered, so we can point to a qubit remapping policy as the sole factor driving performance differences.

We also found that remapping policies perform differently early in training. Table 2 shows the average rankings (lower being better) of qubit remapping policies in early and later stages of training. The greedy minimum entropy works best throughout training, but of note are the relative rankings of maximum correlation and minimum correlation policies. Maximum correlation remapping had an average rank of 2.882 early in training versus minimum correlation’s average ranking of 1.976. However, by epoch 200 the maximum correlation policy’s rank had switched places with the minimum correlation policy’s ranking—1.88 versus 2.290. We felt it important to note this change since the results of Figure 8 show maximum correlation outperforming minimum correlation but Figure 5 shows the opposite. This discrepancy is resolved by Table 2’s split of training into early and late stages.

Based on these results, we performed additional comparisons between remapping policies to see if our results held up across different data sets and when transferred from software to hardware. D-Wave’s software simulator uses bucket-tree elimination to sample thermal distributions [17]. Whereas the results just described were gathered from BMs trained on MNIST digit data on the software simulator, our next set of results were gathered from BMs trained on MoS_2_ data, from running on D-Wave’s annealing hardware, or both. We also remapped qubits every 5 epochs throughout the entirety of training rather than just remapping in epochs 0, 5, 10, and 15. The idea was to increase the influence of remapping policies on final results and widen performance gaps between policies.

Figure 9 shows the results from a BM trained on MoS_2_ data. Overall, the trends observed in Figure 8 remain the same, where low, but not minimal, entropy gives the best results. The main observation here is that the performance of different policies remains the same across both the MNIST digits and MoS_2_ data sets, which strengthens the idea that remapping policies have consistent global and generalized differences. A secondary observation is a noticeable new feature: sudden spikes in error that occur regularly. These performance dips happen every 5 epochs; furthermore, they happen exactly on the epochs when we remap qubits. We expected this result because every remapping action slightly changes the problem we have trained the BM to solve, thus the BM should accordingly perform slightly worse. Recall the discussion of parameter space exploration in Figure 7. In performing a remapping action, we potentially remove a parameter the BM has adjusted and trained upon in favor of another parameter which may have received reduced or no training. The looping, interconnected nature of the BM’s hidden units means the swapping of parameters will negatively impact short-term performance.

To address potential concerns that our experiments were performed on a simulator instead of on annealing hardware, we trained a Boltzmann machine on MoS_2_ data using D-Wave’s quantum annealer.

Figure 10 shows the results gathered from the annealing hardware. We see the same trends as in previous experiments. One noticeable difference is the slightly higher error. This is expected because the hardware has to deal with physical realities such as noise, limited bit precision, and hard limits on parameter ranges [18]. Although these issues are worth discussing in a different context, they are not our focus here. Some constant difference in error numbers aside, our results from the physical hardware align with all of our previous experiments on the software simulator. Overall we have concluded that low entropy, but not minimal entropy, is a good qubit remapping policy that produces consistently better results across data sets in both software and hardware.

## 5. Discussion: Balancing Entropy

Our experimental results suggest an organized method of assigning limited Boltzmann machine hidden variables to nodes in a chimera graph improves performance. The LBM concept and variable reassignment policies can both be implemented on a quantum annealer or used in a classical setting. The approaches of maximizing correlation and minimizing entropy were initially based on intuition. We can explain the motivation behind our approach using the concept of hidden information.

Kamimura et al. [19] examine the entropy of hidden unit activation patterns in autoencoders. Autoencoders produce efficient encodings of data that minimize information loss. The authors put forth the idea that information can be categorized as necessary or unnecessary. Necessary information is intrinsic to the data that being captured in the encodings whereas unnecessary information is random noise or patterns. The goal of the encoding units is to capture necessary information and filter out unnecessary information. As an encoding network trains on input data, the hidden units exhibit low entropy activation patterns. Individual or groups of units begin to specialize as feature detectors that respond specifically to particular data input patterns. Hidden units do not randomly activate in a trained encoding network. If they do behave randomly (high entropy), a network would be unable to capture any meaningful necessary information about the input data. Finding results similar to ours, the authors conclude that low entropy, but not minimal entropy, produces better results in encoding networks.

The amount of hidden information captured by a network is defined as the difference in network entropy at the start of training and the network entropy at the end of training. Using the definition of entropy (Kamimura calculated and summed the individual entropy of every hidden unit whereas in our work we calculated and summed the joint entropy of chimera cell groupings. The general idea and goal remain the same.) in Equation (7), we define hidden information as
(9)$$I=H_{start}-H_{end}$$

This definition of hidden information can be added to the objective function of an autoencoder, giving the network the dual goals of minimizing residuals and maximizing hidden information. Such a gradient would be defined as
(10)$$\frac{\partial I}{\partial w_{jk}}=\sum_{s}^{S}\phi_{j}^{s}\xi_{k}^{s}$$
(11)$$\phi_{j}^{s}=\left(\log P_{j}^{s}-\sum_{r}P_{r}^{s}\log P_{r}^{s}\right)P_{r}^{s}\left(1-h_{j}^{s}\right)$$

Here, *s* is a given input pattern from data set *S*, $\xi_{k}^{s}$ is the *k*th element of input pattern s∈S, and $h_{j}$ is the state of hidden unit *j*. This translates into the weight update rule that includes both hidden information ϕ (with hyper-parameter α) and cross-entropy δ (with hyperparameter β) measures:(12)$$\Delta w_{ij}=\sum_{s}^{S}\left(\alpha \phi_{j}^{s}+\beta \delta_{j}^{s}\right)\xi_{k}^{s}$$
(13)$$\delta_{j}^{s}=f^{\prime }\left(u_{j}^{s}\right)\sum_{i}^{N}w_{ij}\delta_{i}^{s}$$
where $u_{j}^{s}$ is the activity level of hidden unit *j* (so just $\sum_{k}^{L}w_{jk}\xi_{k}^{s})$, and
(14)$$\delta_{i}^{s}=\xi_{i}^{s}-O_{i}^{s}$$
where $O_{i}^{s}$ is the target output of the *i*th unit of pattern s∈S. When using an encoding network, $O_{i}^{s}$ is original input data.

Networks that utilize Equation (12) tend to perform better than standard networks. However, the balance of the $\frac{\alpha }{\beta }$ ratio has significant effects on network performance, where too much weighting of α (hidden information and entropy) causes overall results to suffer.

In our problem with BMs, we search for subgroup pairings that produce low entropy. Supposing we have two subgroups *L* (left) and *R* (right), we can be in one of three situations. In the first, R=f(L), a minimal entropy arrangement. That is, *R* is simply a function of *L*. Then, it follows that P(L,R)=P(R|L)P(L)=P(L) and H(L,R)=H(L). However, our procedure is unlikely to find such a pairing, if it even exists, due to the random nature of weight initialization and the distribution of our data.

The opposing extreme situation is one where *L* and *R* are independent and produce maximum entropy, or H(L,R)=H(L)+H(R). Our procedure is unlikely to pick this arrangement because we seek low entropy.

We are most likely to find ourselves in the third situation, where we are between the two extremes. For every state vector $L=(l_{0},\ldots l_{n}-1)$, *R* can return one of *k* different state vectors. Then H(L,R)=H(R|L)+H(L)=log(k)+H(L), where $k=2^{4}$ at most in our case where we have 4-unit subgroups. The intuition behind this situation is that our procedure finds an *R* that returns a small number of patterns. This has the effect of making our hidden units respond selectively to input.

While our work differs from Kamimura et al.’s, there are enough similarities that the same principles apply. Although Kamimura used deterministic autoencoders to examine entropy and hidden information, probabilistic Boltzmann machines perform largely the same encoding function—when exposed to input data, a set of stochastic hidden units is activated in response. When applying weight update rules, Kamimura explicitly includes a hidden information term whereas we implicitly use it by choosing to include or drop certain hidden-hidden connections based on entropy measures. Both approaches try to minimize entropy in hidden unit encodings and both see improved results.

We have seen that entropy has a significant effect on our experimental results and that we can manage entropy via the choices we make in our variable-to-qubit assignment decisions. A principle we have observed is that low entropy, but not minimal entropy, is generally desirable. A question that naturally arises is: what level of entropy in our qubit mapping is optimal for LBM performance?

This question has parallels to existing problems in machine learning and statistics. Many learning or optimization problems include a regularization term in their loss function. The regularization term generally acts as a balance against some specific form of misbehavior by the model, be it complexity, overfitting, saturation, or some other offense. Therefore, supposing a loss function V(f(x),y) for the output of f(x) where the label is *y* and a regularization term R(f) that penalizes some behavior by f(x) itself, we have a loss function L(x,y):(15)$$L\left(x_{i},y_{i}\right)=V\left(f\left(x_{i}\right),y_{i}\right)+\lambda R\left(f\right)$$

Though the exact form of *R* varies according to specific needs, determining the proper value of the λ term is typically achieved through an empirical tuning process [20,21,22,23]. As it stands, our approach separates entropy from the regular LBM training and weight update process into two steps: we first update network parameters normally, without consideration of entropy, then evaluate the entropy of our mapping and potentially reassign variables to qubits. We believe it possible to treat entropy as a regularization term and directly include it in the weight update process, explicitly tuning α to find good qubit mappings.

## 6. Conclusions

While most BMs use a restricted topology to make training tractable, quantum annealers allow us to ease topology restrictions and include hidden-to-hidden connections. Our experiments using these LBMs have shown improved performance over RBMs.

The LBM design was created with quantum annealers in mind. Though adding connectivity back into BMs is conceptually straightforward, implementing LBMs on an annealer is difficult due to limited qubits and sparse hardware connectivity. We fit LBMs onto annealers by representing only the hidden units using qubits and the intra-layer connections using couplers.

Because the connectivity of LBMs leads to intractability issues during training, we use the sampling properties of an open-system quantum annealer to estimate state distributions in training. Our approach also avoids expensive graph embedding problems. Rather than designing an predetermined connectivity topology for our LBMs and fitting a sparse annealer hardware topology to it, we instead use the annealer topology “as is” and manipulate topology through qubit mapping instead.

We found that mappings maximizing correlation between qubit activity within chimera cells consistently produced better results than the hand-crafted mappings we used in previous experiments. We extended our approach by using low entropy as a metric to design qubit mappings.

Future work may consider how to adapt these findings to new generations of hardware, such as the Pegasus architecture announced by D-Wave that features 15 connections per qubit and 5000 qubits [24]. Our work does not rely on a specific number of qubits or couplers, so it can likely be generalized to handle the introduction of new machines. More qubits and couplers would also allow us to build richer, multi-layered networks. Our experiments with Boltzmann machines had assumed a single layer of hidden units due to size constraints, so a larger machine could enable us to experiment with deep Boltzmann machines with multiple layers of hidden units.

## Figures and Tables

**Figure 1 entropy-22-01202-f001:**
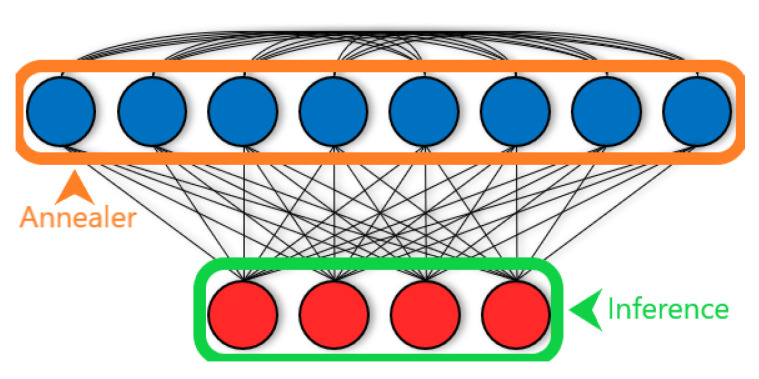
A limited Boltzmann machine (LBM). Connectivity within a LBM is a superset of connectivity within a RBM. In addition to all the bipartite hidden (blue) to visible (red) connections allowed to RBMs, LBMs also allow connections to exist between hidden units. Given a M×N grid of chimera cells, there are 16MN+4M(N−1)+4N(M−1)=O(MN) connections within the LBM we implement on D-Wave’s quantum annealer. To carry out contrastive divergence training, we need to find P(v|h) (green) and P(h|v) (orange) using Gibbs sampling. We can easily infer distributions in RBMs—due to conditional independence induced by the bipartite connectivity structure, the distribution of *v* factorizes when we fix the states of *h*, and vice versa. However, for LBMs, the connections among hidden units means P(h|v) no longer factorizes and we cannot exactly infer the distribution. Instead, we program a quantum annealer to represent the LBM problem and draw samples to estimate P(h|v). In summary, in LBMs we can analytically find P(v|h) but we estimate P(h|v) using a quantum annealer. Knowing both quantities allows us to train the LBM using Equation (3).

**Figure 2 entropy-22-01202-f002:**
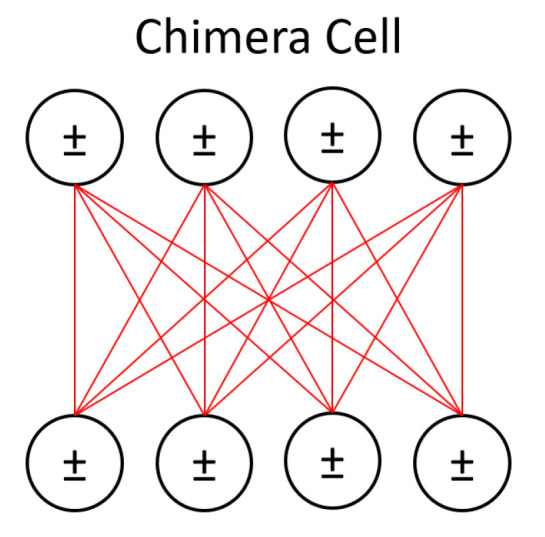
A portion of the hidden layer from Figure 1 is represented using one of D-Wave’s chimera cells here, with the cell’s bipartite connectivity made more obvious. The visible units of the LBM are left on a classical machine while all qubits in this figure represent hidden units. The effect of visible units on the activation of hidden units is reduced to an activity bias (represented with ± symbols) on those units.

**Figure 3 entropy-22-01202-f003:**
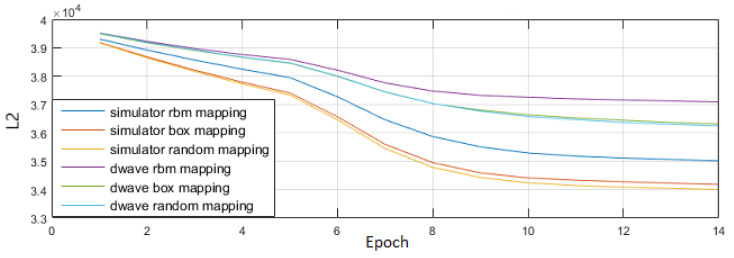
A comparison of different qubit mapping methods on both software simulator and the D-Wave processor. Vertical axis is total reconstruction error over the MNIST data set and horizontal axis is training epoch. A plain RBM run on each platform is shown as baseline and are the weakest performers. The initial qubit-mapping scheme was the “box” mapping where qubits that cover adjacent pixels were grouped together in chimera cells. Some random mapping, featuring mixes of spatially adjacent qubits and more global connections, outperform the box mapping.

**Figure 4 entropy-22-01202-f004:**
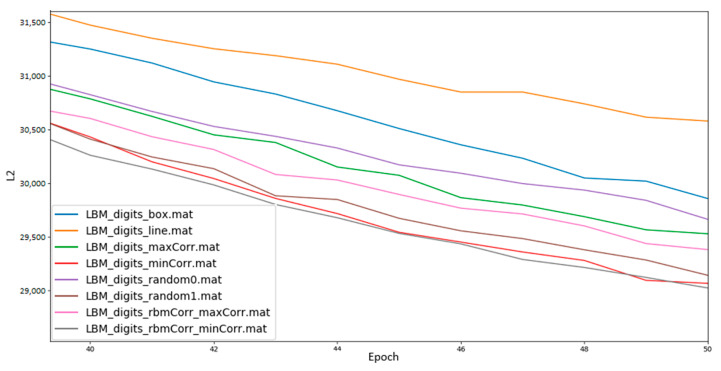
Terminal results of LBMs using qubit mappings generated by varying correlation within chimera cells. We include our previous box and line mappings as comparisons. The maxCorr series results from maximizing correlation in the mapping and minCorr series from minimizing correlation. As noted, some random mappings were able to outperform those box and line mappings, and two such random mappings are shown. Results labeled *rbmCorr* are produced via RBM-to-LBM conversion and correspondingly by minimizing (minCorr) or maximizing (maxCorr) correlation.

**Figure 5 entropy-22-01202-f005:**
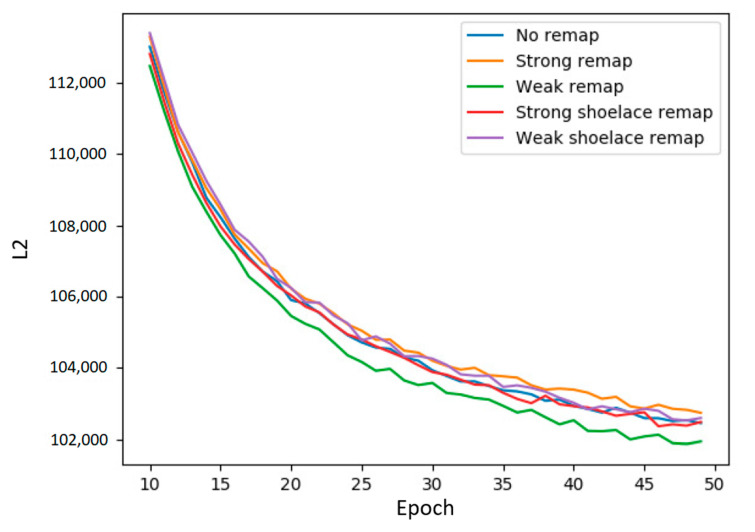
A comparison of different policies on the MoS_2_ data set. The y-axis is reconstruction error and the x-axis is training epochs. Our original policy is the “weak” remapping that places uncorrelated qubits together in the same chimera cell. The opposite policy, the “strong” remapping, places highly correlated qubits together. The variant “shoelace” policies achieve the same aims albeit in a slightly different manner. As shown, the results on the new data set align with our initial results on the MNIST digits data set. When trained for less than 50 epochs, our original “weak” policy is the best performer, the opposite “strong” policy the worst, and all other policies (including choosing not to remap at all) are in the middle.

**Figure 6 entropy-22-01202-f006:**
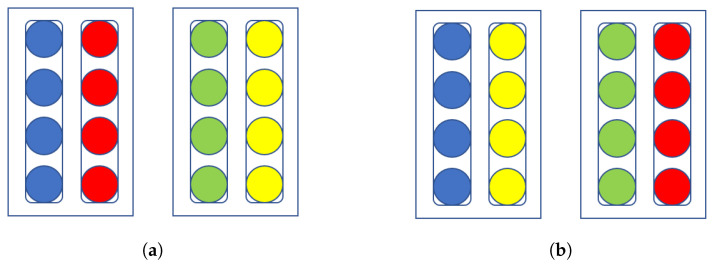
A visual color-coded description of how we remap logical qubits to hardware qubits. (**a**) The hidden units of a Boltzmann machine mapped onto the chimera cells of the D-Wave quantum annealer. Here we have a simplified network containing only 2 chimera cells. Note the subgroups, represented by the rounded boxes, are composed of 4 units and that each chimera cell, represented by the angular boxes, contains 2 such subgroups/partitions. (**b**) The same network but with subgroups/partitions swapped. The qubits have new neighbors to interact with and new entropy values to calculate.

**Figure 7 entropy-22-01202-f007:**
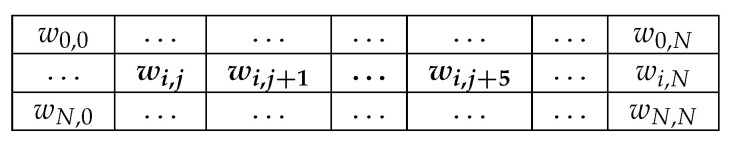
The parameter space qubit *i* is allowed to explore. Shown is a matrix of weight values. Qubit *i*’s connectivity is limited to merely 6 other qubits, which we conveniently listed as qubits *j* through j+1, so the possible space is quite small. Only the bolded parameters can be changed under a static mapping method, whereas a dynamic mapping method can alter any weight parameter $w_{ij}$ such that i<j.

**Figure 8 entropy-22-01202-f008:**
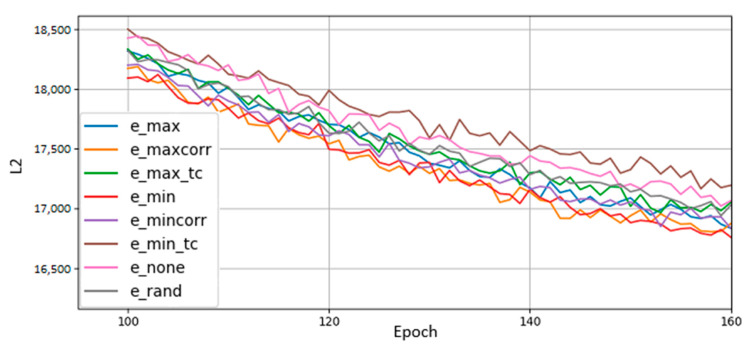
Comparison of qubit mapping policies. The greedy policy (e_min) performs best overall, achieving a loss value at epoch 160 that a non-remapped BM needs 200 epochs to reach, cutting off 25% of training time. Total correlation policies (e_min_tc, e_max_tc) are also shown, though they did not stand out in particular. Random (e_rand) and 1:1 (e_none) mappings are included for comparison, along with the correlation policies from before (e_mincorr, e_maxcorr).

**Figure 9 entropy-22-01202-f009:**
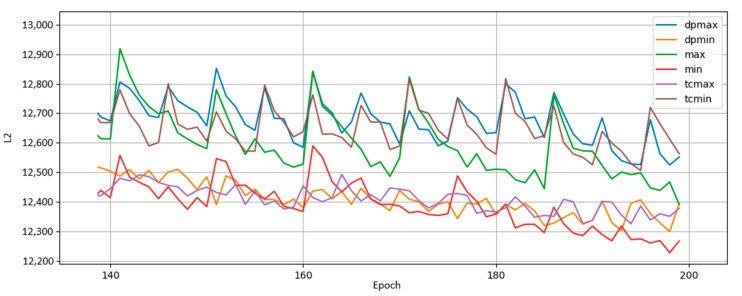
Results from a BM trained on MoS_2_ data run where qubits are remapped every 5 epochs throughout training. The policy rankings (greedy policies “min” and “max”, greedier policies “dpmin” and “dpmax”, and total correlation policies “tcmin” and “tcmax”) hold steady across data sets, suggesting the usage of entropy as a metric for remapping decisions may be a generally good idea regardless of the input data. The upward spikes of L2 occur every 5 epochs upon remapping.

**Figure 10 entropy-22-01202-f010:**
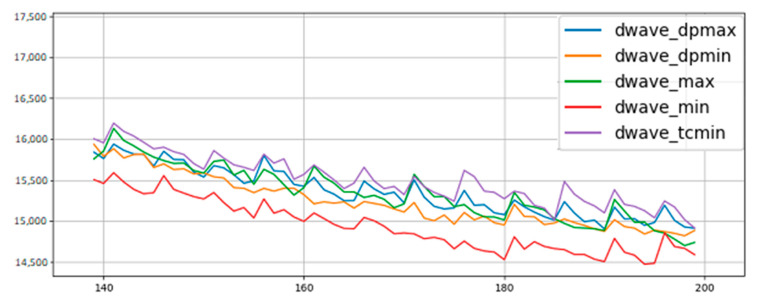
BMs trained on MoS_2_ data using annealing hardware. Policy trends remain the same as before, though the greedy minimum entropy policy stands out more as a better performer.

**Table 1 entropy-22-01202-t001:** A table of entropy values for a given mapping. We compare our greedy minimal mapping method against our greedier minimum method using the MNIST data set. We see that the greedier method produces lower entropy values. Remapping occurs every 5 epochs.

Remap	Greedy Minimum	Greedier Minimum
0	79.50	79.18
1	78.15	75.89
2	72.95	69.21
3	70.14	66.04

**Table 2 entropy-22-01202-t002:** Rank statistics on the relative performance of each policy. Table shows average rank for early training epochs (15–100), and average rank for late training epochs (100–200); this BM was trained on the MNIST digits data set using D-Wave’s software simulator. “TC” stands for total correlation. The minimum correlation policy initially works better than the maximum correlation policy, consistent with our results in Figure 5, but then switches rank in relative performance later in training.

Policy	Rank, Early Epochs	Rank, Late Epochs
Greedy Max Entropy	3.435	3.005
Max Correlation	2.882	1.88
Max TC	4.200	4.245
Greedy Min Entropy	**0.317**	**0.675**
Min Correlation	1.976	2.290
Min TC	4.365	5.605
No Remapping	5.845	6.282
Random	4.541	4.455

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
