# Peer review of "Dynamic Topology Reconfiguration of Boltzmann Machines on Quantum Annealers"

_entropy, 2020, doi:10.3390/e22111202_

Round 1
Reviewer 1 Report
The authors propose and experimentally test methods for using quantum annealers as Boltzmann machines. They use standard data sets and D-Wave quantum annealers to perform these tests. The novel element of their method is to use tools which re-map the qubits to take maximum advantage of the limited connectivity. While I am not an expert on Boltzmann machines, the methods do seem reasonable to me, and the results are interesting. Assuming the machine learning aspects of the methods are sound (I do not feel fully qualified to make this judgment), then I think this paper should be published with only relatively minor modifications.
While I do feel that the authors have made their methods relatively clear, making at least examples of the code which they used publicly available would be useful (assuming they are not prevented by funder or other restrictions). Ideally it would be best for all of the code to be made available, but if this is not practical, it should be at least enough to reproduce the basic experiments (assuming access to a D-Wave or a simulator of one). In particular, making the experimental code available could increase reproducibility by making it clear exactly what methods were used, for instance if spin inversions (aka gauge averaging) were used.
Minor comments:
The authors are very clear about the open system nature of the D-Wave quantum annealer, which is a mark in their favour but the phrase "open system adiabtic" is a bit of a contradiction in terms since adiabatic literally means without heat transfer, I would recommend instead saying "open system quantum annealer"
The authors should clarify the discussion on line 268 where they discuss the likilihood of adding more chimera cells with later discussion of the new topology, I appreaciate that chimera cells are subgraphs of the new Pegasus archetectures, but many readers will not know this.
line 286, "have find" should be "have found"
line 354 "only only"
line 454 Please explain which software simulator is being used here and outline the methods which it uses if this is the 4x4 chimera exact solver which comes packaged as part of the solver, than I believe it uses a techique known as bucket tree elimination [Kask, K., Dechter, R., Larrosa, J. & Fabio, G. "Bucket-tree elimination for automated reasoning" Artif. Intel. 125, 91–131 (2001)] which is an exact method for sampling thermal distributions, but the authors should reach out to the D-Wave support team to confirm.
When discussing precision effects from running on the real device at the end of section 4, the authors should probably reference [Chancellor et. al. https://doi.org/10.1038/srep22318] which shows that this kind of error does indeed have a major effect on thermal sampling applications
Author Response
Please see attachment. We can also provide code examples, but they were written for the old sapi

Reviewer 2 Report
The authors propose to use a quantum annealer as a sampling module for a machine learning procedure called the Boltzmann Machine. Training the machine requires knowing averages of some variables over some complicated distributions. Calculating these distributions and sampling from them is in general inefficient for standard methods. People deal with the complexity by simplifying the machines (Restricted Boltzmann Machines). It is reasonable to believe that more complicated machines will lead to better results. Therefore, it is reasonable to look for sampling devices that allow us to sample from more complicated distributions. In recent years, quantum annealers were proposed to play the role of the samplers. Among the challenges of this solution we have the limited number of nodes and the fixed connectivity between nodes in the hardware. The proposals of using the annealers as samplers for the Boltzmann Machines do not appear in this manuscript for the first time. However, this manuscript proposes some solutions for the challenges of the hardware. The new things proposed here are: to implement on the hardware only part of nodes of the machine (the so-called hidden nodes); to introduce some connections between the hidden nodes (Limited Boltzmann Machines); to associate bits of data with the nodes according to some rules implied by correlations or information theory. The numerical studies of different approaches are here completed by an experiment using a D-wave quantum annealer.
I find this research an interesting and important contribution to current research regarding the application of quantum computers or simulators to practical tasks, specifically machine learning tasks.
There are discussions whether the quantum annealing as suggested in this manuscript is more efficient than the best classical algorithm. However, in my opinion, regardless the answer this work provides a useful contribution as a bridge between machine learning procedures and quantum or quantum inspired algorithms.
The only improvement I would suggest is to provide some kind of schematic summaries of the contrastive divergence procedure and the procedure of sampling from a quantum annealer according to a given distribution. I think that specialists know these details, but they could be useful for other readers.
I support publication of this research.
Reviewer 3 Report
The authors state that they have applied an open-system adiabatic quantum annealer and implemented Boltzmann machines with looping connectivity. They also mentioned that they have improved the performance.
The subject is interesting however before giving my final decision, some critical remarks should be answered and taken into consideration for the revised version:
- The authors studied the correlations without identified if it is classical or not? This should be clearly stated.
- The author should introduce a brief description of the D wave processor.
- The authors omit to introduce several concepts briefly such as correlation and quantum correlations. The authors should write an extended paragraph on the quantum correlations in physical systems this will increase the visibility of the paper. The following references are helpful and should be cited: PRA 91, 032309 (2015); Quant. Inf. 12, 69 (2013); PRA 84, 053817 (2011), and references therein.
- The authors should clarify more in details the derivation of eq 3, references also should be added.
- The authors with an open system. Discussion on the effect of the environment and alternative description of open systems should be briefly discussed. The references EPJD 69, 229 (2015) and PRA 93, 042116 (2016) as well as references therein are helpful.
Round 2
Reviewer 3 Report
The paper needs a second round of revision.
The response and the revision of the papers still not satisfying the most of the previous comments.